# Phytochemicals and Their Possible Mechanisms in Managing COVID-19 and Diabetes

Eli Mireya Sandoval-Gallegos [1], Esther Ramírez-Moreno [1], Nancy Vargas-Mendoza [2], José Arias-Rico [3], Diego Estrada-Luna [3], José Javier Cuevas-Cancino [3], Reyna Cristina Jiménez-Sánchez [3], Olga Rocío Flores-Chávez [3], Rosa María Baltazar-Téllez [3] and José A. Morales-González [2],*

[1] Área Académica de Nutrición, Centro de Investigación Interdisciplinario, Instituto de Ciencias de la Salud, Universidad Autónoma del Estado de Hidalgo, Circuito Ex Hacienda La Concepción S/N, Carretera Pachuca-Actopan, San Agustín Tlaxiaca 42160, Mexico; eli_sandoval7987@uaeh.edu.mx (E.M.S.-G.); esther_ramirez@uaeh.edu.mx (E.R.-M.)

[2] Laboratorio de Medicina de Conservación, Escuela Superior de Medicina, Instituto Politécnico Nacional, México Escuela Superior de Medicina, Plan de San Luis y Díaz Mirón, Col. Casco de Santo Tomás, Alcaldía Miguel Hidalgo, México City 11340, Mexico; nvargas_mendoza@hotmail.com

[3] Área Académica de Enfermería, Instituto de Ciencias de la Salud, Universidad Autónoma del Estado Hidalgo, Circuito Ex Hacienda La Concepción S/N, Carretera Pachuca-Actopan, San Agustín Tlaxiaca 42160, Mexico; jose_arias@uaeh.edu.mx (J.A.-R.); destrada_luna@uaeh.edu.mx (D.E.-L.); jose_cuevas@uaeh.edu.mx (J.J.C.-C.); jimenezs@uaeh.edu.mx (R.C.J.-S.); ofloresc@uaeh.edu.mx (O.R.F.-C.); rosa_baltazar@uaeh.edu.mx (R.M.B.-T.)

* Correspondence: jmorales101@yahoo.com.mx

**Abstract:** For the writing of this manuscript, we searched information published from 2000 to 2021, through PubMed, Web of Science, Springer, and Science Direct. Focusing on the effects related to respiratory diseases, in addition to possible direct effects towards SARS-CoV-2, coupled with diabetes. Diabetes is a metabolic disease that is characterized by affecting the function of glucose, in addition to insulin insufficiency. This leads to patients with such pathologies as being at greater risk for developing multiple complications and increase exposure to viruses infections. This is the case of severe acute respiratory disease coronavirus 19 (SARS-CoV-2), which gave rise to coronavirus disease 2019 (COVID-19), declared an international public health emergency in March of 2020 Currently, several strategies have been applied in order to prevent the majority of the consequences of COVID-19, especially in patients with chronic diseases such as diabetes. Among the possible treatment options, we found that the use of phytochemical compounds has exhibited beneficial effects for the prevention and inhibition of infection by SARS-CoV-2, as well as for the improvement of the manifestations of diabetes.

**Keywords:** diabetes; SARS-CoV-2; COVID-19; curcumin; silymarin; sulforaphane

## 1. Introduction

At present, humanity finds itself in the presence of a virus that has placed the life of the population in danger. The severe acute respiratory syndrome (SARS), better known as the SARS-coronavirus 2 (CoV-2) virus, which was discovered recently, is the cause of the infectious disease denominated coronavirus disease 2019 (COVID-19), identified for the first time in Wuhan, China. According to the monitoring carried out by the World Health Organization (WHO), globally, up to 30 July 2021, 1,996,553,009 new cases of COVID-19 had been reported, in addition to 4,200,412 deaths [1].

The population infected by COVID-19 exhibited the development of respiratory symptoms, with those ranging from slight to moderate recovering without basic treatment against this disease. However, COVID-19 can become severe in persons presenting underlying diseases, such as diabetes or some other pathologies, which cause the development of serious symptoms [2].

Specifically, in the case of patients with diabetes infected by SARS-CoV-2, investigations have determined that the latter compromises the immunological system, which provides protection against any damaging event to the organism, in addition to originating an excessive proinflammatory cytokine storm, which, in turn, gives rise to the acute respiratory distress syndrome (ARDS). On the other hand, it has been demonstrated that the SARS-CoV-2 virus causes direct effects on pancreas [3]. Exhaustive investigations have been conducted in the search for novel alternatives, among which are the use of phytochemical compounds that, in agreement with the literature, tend to be promising, not only against SARS-CoV-2, but also as coadjuvants in diabetes. In terms of methodology used, research was obtained through PubMed, Web of Science, Springer, and Science Direct. SARS-CoV-2; acute respiratory distress syndrome (ARDS); and curcumin, silymarin, and sulforaphane related with respiratory diseases were set as keywords to search for relevant studies between the years 2000–2021.

## 2. Diabetes

Diabetes is a metabolic disorder that affects the function of glucose in the human body due to insulin deficiency or to its own actions [4]. According to estimates performed by the WHO, it has been indicated that, for the year 2014, 8.5% of the adult population had diabetes, and that by 2016, it was the direct cause of 1.6 million deaths. It has been noted that one of every 11 persons develops diabetes and, by 2025, it is estimated that 380 million persons will develop diabetes [5,6].

There are two possible causes for developing diabetes. It may be due to low or nil insulin functioning in the metabolism, mainly relating to glucose; this may be because of the increase of the intracellular concentration of the fatty acids metabolite, which activates a serum kinase cascade, triggering defects in insulin signaling due to the diminution of the insulin receptor [7]. On the other hand, the second cause of diabetes can be due to the lack of insulin production in the body, the latter because of the destruction of the β-cells of the Langerhans islets in the pancreas; this has been referred as a triggering mechanism, due to the interaction between the pancreatic β-cells and the innate and adaptative immune systems [8]. DiMeglio et al. [9] referred to the lack of insulin as the cause of the exposure of the β-cell peptides to the antigen-presenting cells (APC). In this, the autoantigens are displaced to the pancreatic lymph glands by the APC, where they interact with the autoreactive CD4+ T lymphocytes, in turn mediating the activation of the autoreactive CD8+ T lymphocytes, in which the activated CD8+ T cells return to the islet and lyse the β cells. Also, destruction of the β cells is aggravated by the release of proinflammatory cytokines and the reactive oxygen species (ROS) of the innate immune cells. Similarly, the activated T cells within the pancreatic lymph nodes also stimulate the B lymphocytes in order for these to produce antibodies against β-cell proteins.

*Respiratory Infections in Diabetes*

Poorly controlled or decompensated patients with diabetes are at greater risk of contracting infections in any organ or tissue [6,10]. Affectations of the respiratory system are very frequent because the hyperglycemic environment renders the individual more susceptible, this is considered as a potential independent factor in the development of infections of the lower respiratory tract (Figure 1). However, there are other factors that trigger respiratory events, such as advanced age, unhealthy lifestyle, diminished immunological function (damage to neutrophil and macrophage function, depression of the antioxidant system, humoral immunity, phagocytosis, and chemotaxis), a greater risk of aspiration due to diabetic gastroparesia, a possibly deteriorated pulmonary function, and pulmonary microangiopathy, among other factors [6,11–13]. In this manner, the diabetic population is predisposed to the triggering of pulmonary infections caused by metabolic alteration due to diabetes and immunosuppression [12].

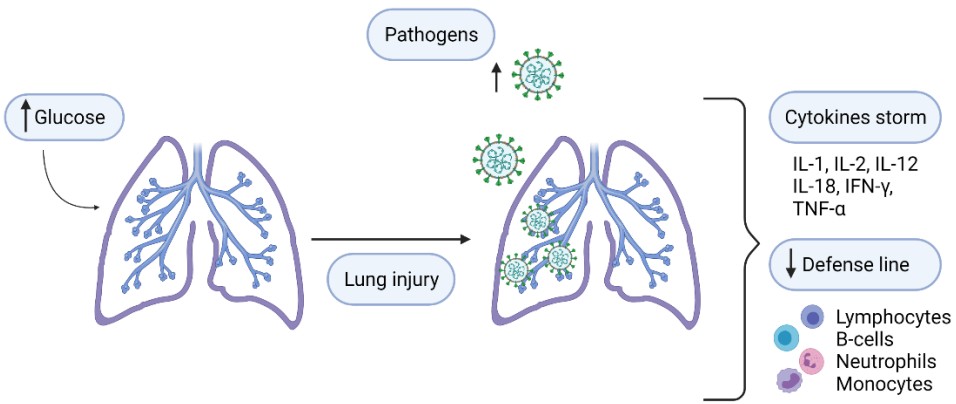

**Figure 1.** Hyperglycemia in the lungs. The increase of glucose in the lungs causes lesions and the inflammatory state, which in turn provokes the line of defense in the lungs for diminishing and giving rise to an increase of pathogens, in addition to pulmonary damage. Created with BioRender.com (accessed on 1 September 2021).

## 3. COVID-19

COVID-19 is a serious respiratory disease associated with pneumonia, produced by SARS-CoV-2, a β-coronavirus denominated as a highly contagious pathogen. It has been observed that SARS-CoV-2 is transmitted through respiratory droplets expelled principally from one person to another. Symptoms appear, on average, 5 days after exposure; however, these could appear at up to 11.5 days [14]. SARS-CoV-2 is a virus that utilizes the angiotensin-3-converting enzyme (ACE2) as a cellular receptor in humans, causing pulmonary interstitial damage and even provoking parenchymatous changes [15–17]. The clinical manifestations of COVID-19 are very diverse; nevertheless, the most common of these are fever, cough, and dysnea [18]. On the other hand, according to the severity of COVID-19, it has been classified in three levels as follows: slight (fever, cough, fatigue, with pneumonia or with slight pneumonia), severe (dysnea, blood oxygen saturation of ≥93%, respiratory frequency ≥30/min, the relation of the partial pressure of arterial oxygen at an inspired oxygen fraction of >300, pulmonary infiltrates of >50% within 24–48 h, and the need for being in the intensive care unit [(ICU)), and critical (acute respiratory distress syndrome ([ARDS]), including respiratory insufficiency, septic shock, and/or dysfunction or multiple organ insufficiency; metabolic acidosis; and coagulation dysfunction) [19].

*Diabetes and COVID-19*

According to previous reports, the angiotensin–renin system (ACE) functions as a regulator in diverse pathologies, among which are found cardiovascular pathologies, hypertension, kidney disease, diabetes, and pulmonary diseases. Prior investigations have suggested that SARS-CoV-2 couples, to a great degree, with the augmented expression of the angiotensin-converting enzyme 2 (ACE2) in metabolic organs and tissues, in type 2 alveolar cells (AT2), myocardial cells, pancreatic beta cells (β-cells), adipose tissue, small intestine, and kidneys, favoring a greater cellular binding of SARS-CoV-2 [20,21]. Recently, it has been suggested that the SARS-CoV-2 virus can bind to the cells through the ACE2 [22]. In the presence of acute hyperglycemia the expression of ACE2 increases in cells, permitting entry of the virus. In chronic hyperglycemia, there is a low regulation of ACE2 expression; nonetheless, the cell continues to be exposed to a major inflammatory effect and to damage induced by the virus [23]. Das et al. [24] mentioned that SARS-CoV-2 aggravates the situation in the patient with elevated glucose levels, possibly the consequence of the increased response of proinflammatory substances, due to the inefficiency of the innate immunity and of the negatively regulated ACE2. Additionally, it has been found that diabetes is related to the expression of ACE2 in the lung. Muniyappa et al. [21] mentioned the following mechanisms by which SARS-CoV-2 infection in diabetic population can increase: (1) cellular binding of greater affinity and efficient entry of the virus, (2) diminution of viral clarifica-

tion, (3) diminution of T-cell function, (4) increase in susceptibility to hyperinflammation and to the cytokine storm syndrome, and (5) the presence of cardiovascular diseases (CVD). Codo et al. [25] suggested that once infected with the the virus (lungs), the monocytes and macrophages that have been often observed in the lungs of patients with COVID-19 and in those with high concentrations of glucose, adapt their metabolism through the infection and become highly glycolytic, facilitating the replication of SARS-CoV-2. On the other hand, mitochondrial reactive oxygen species (mtROS) are produced, these molecules inducing stabilization by means of the hypoxia-inducible factor 1 (HIF-1a), promoting glycolysis. Thus, the modifications induced by HIF-1 in the metabolism of the monocytes due to the presence of SARS-CoV-2 inhibit the response of the T cells and promote the death of pulmonary epithelial cells.

The use of ACE inhibitors or angiotensin receptor blockers has been suggested as a part of the therapy [26]; however, this would not be completely recommended, because it would alter the glucose level in patients with COVID-19 infection and generate other affectations, such as cell damage, hypopotassemia, and the increase of cytokine and fetuin-A insulin resistance, with such conditions possibly worsening in COVID-19 [24]. On the other hand, the ACE inhibitors and angiotensin receptor blockers (ARB) can trigger a greater expression of ACE2, allowing viral uptake, and can intensify the risk of serious infection in individuals with diabetes [27]. The results obtained by Codo et al. [25] considered the existence of mtROS/hypoxia-inducible factor 1 (HIF-1)/glycolysis as a target in order to develop treatments that participate as coadjuvants for improving disease due to COVID-19.

## 4. Alternative Treatments with Phytochemicals (Curcumin, Silymarin, and Sulphorafane)

There are several compounds that have been identified as having hypoglycemic and antiviral activities. However, the phytochemicals curcumin, silymarin, and sulforaphane have been widely used in world population, either as elements of the diet or in traditional medicine to treat medical conditions. Recently, these compounds had taken scientific attention because of the different properties described, such as anti-inflammatory, antioxidant, antimicrobial, and hepatoprotective properties, among others.

### 4.1. Curcumin

Curcumin is a polyphenolic compound, obtained from the rhizome of the plant *Curcuma longa* and *curcuma* spp. of the Zingiberaceae family of Asian origin [28,29], with its use extending from India to China [30,31]. Curcumin is employed in gastronomy and in food and pharmaceutical industries; however, it has been utilized in traditional medicine due to its widely described therapeutic benefits (antiseptic, analgesic, anti-inflammatory, antimalarial, as an insect repellent, etc.). On the other hand, there are reports on curcumin with respect to it maintaining beneficial biological and pharmacological effects, such as antioxidant, anti-inflammatory, cardioprotector, antimicrobial, nephroprotector, hepatoprotector, immunomodulatory, hypoglycemic, hyperlipidemic, and antirheumatic effects [32]. For these reasons, curcumin improves inflammatory alterations occasioned by chronic obstructive pulmonary disease (COPD), acute respiratory distress syndrome (ARDS), pulmonary fibrosis, acute pulmonary lesion, and in lung cancer [33].

Several studies have marked curcumin as an effective therapeutic compound in many diseases. However, it has been shown to have lower bioavailability due to the low aqueous solubility oral [34,35]. To improve the bioavailability of curcumin, numerous approaches have been undertaken. These approaches have involved, firstly, the use of an adjuvant, like piperine, which interferes with glucuronidation; secondly, the use of liposomal curcumin; thirdly, curcumin nanoparticles; fourthly, the use of curcumin phospholipid complex; and fifthly, the use of structural analogues of curcumin [35–37].

Curcumin and Its Effect on COVID-19 and Diabetes

Currently, in the face of the demand for effective treatments for patients with diabetes and infected by SARS-CoV-2, the use of phytochemical compounds has been suggested,

which could involve a coadjuvant option in the therapy of these patients. As reported by Tabatabaei-Malazy [38], curcumin could act as an effective antiviral, possessing functions such as inhibition of the virus, blocking 3-chymotrypsin-like cysteine protease (the 3CLpro enzyme controls coronavirus replication and is necessary in the life cycle of the virus), the signaling of nuclear factor-kappa beta (NF-kB), and the inhibition of the production of proinflammatory markers. It has been suggested that curcumin could possess the ability to impair the entry of the virus into the cell, restraining encapsulation and inhibiting viral protease, in addition to regulating different signaling pathways [39].

On the other hand, curcumin is proposed as an anti-inflammatory and immunomodulatory agent, with an antifibrotic and pulmonary effect. This is in addition to it inhibiting NF-kB and several inflammatory cytokines, as well as inflammatory enzymes such as COX2 and iNOS (which tend to increase in serious cases of SARS-CoV-2) [39]. In addition, it inhibits the expression of the proinflammatory enzyme 5-LOX, as well as chemokines, and it reduces the expression of the C-reactive protein (CRP). Alternatively, curcumin expresses Nrf2 (transcription nuclear factor erythropoietic-2-related factor-2), which inhibits the inflammasome, and, consequently, the development of inflammation, as well as cellular damage in injured organs [40]. Also, the activation of Nrf2 by curcumin promotes the biological effects through interaction with Cys151 in Keap1, inhibiting the pathologies in which oxidative stress, such as type 2 diabetes and cardiovascular diseases and various infections, are involved. Thus, it has been proposed that curcumin can be a therapeutic agent against SARS-CoV-2 [41] (Figure 2).

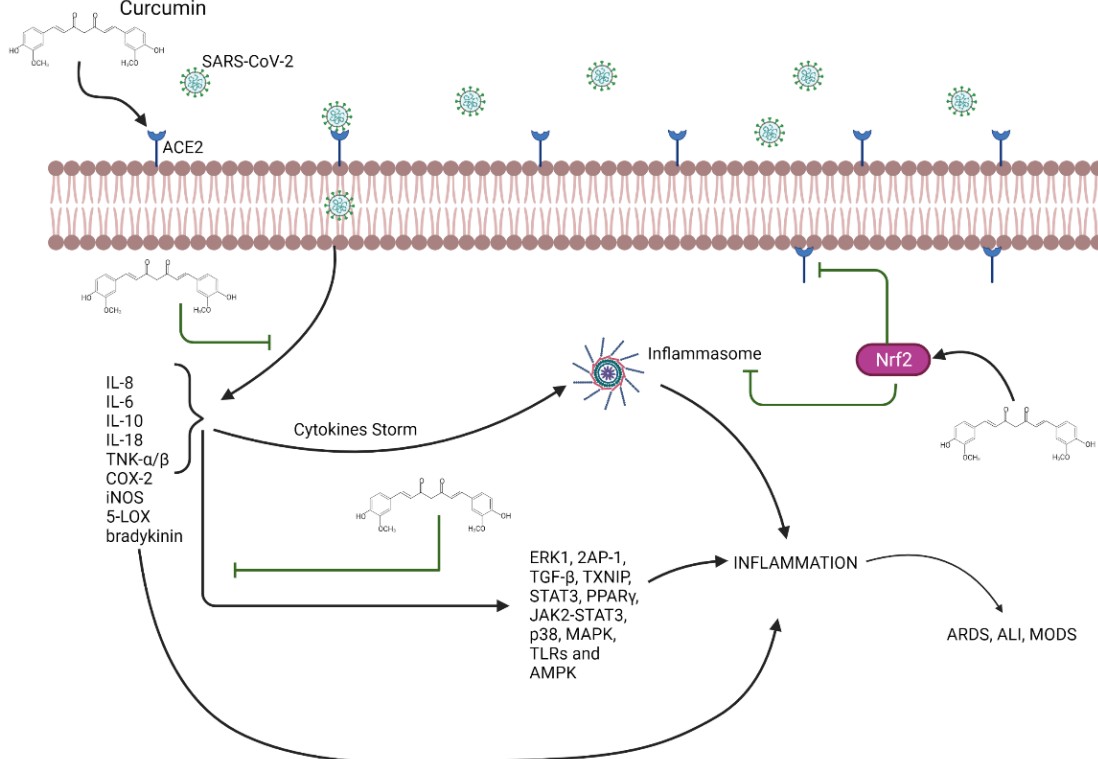

**Figure 2.** The mechanism of action of curcumin. Curcumin can act through the possible interaction that it entertains with the S protein of the virus, blocking the entry of the virus into the cell. Curcumin inhibits exposure to proinflammatory cyto-kines, such as inflammatory enzymes. In addition to blocking transduction pathways, it blocks activation of the in-flammasome and inhibits the expression of ACE2 through the activation of Nrf2 caused by curcumin. Therefore, cur-cumin inhibits inflammation and avoids the development of the acute respiratory distress syndrome (ARDS), Acute Lung Injury (ALI), and the Multiple Organ Dysfunction Syndromes (MOD) of lungs, liver, kidneys, and brain. Created with BioRender.com (accessed on 1 September 2021).

According to the activity that curcumin presents, it has been suggested that it could inhibit the production of inflammatory cytokines, TGF-β collagen, TN-C, α-SMA and E-cadherin, and NF-kB signaling COX-2, as well as Nrf2, PPARγ, JNK, and P38 in severe viral pneumonia, besides inhibiting fibrosis, in addition to activating the nuclear factor erythropoietic-2-related factor-2 (Nrf2)/heme oxygenase 1 (HO-1) signaling axis, which reduces the oxidative stress caused by a viral environment [42]. Investigations in silico have suggested that curcumin is a promising bioactive compound against COVID-19, due to the fact that it has been proposed that curcumin can bind the spike S (glycoprotein that forms homotrimers prominent on the viral surface) of the SARS-CoV-2 virus [43–45]. Exhaustive investigations conducted on the effect of curcumin on SARS-CoV-2 have suggested that this molecule and one of its derivatives could bind with Mpro (Mpro protease), indispensable for the maturation of SARS-CoV-2 [45]. Additionally, curcumin could bind to the glycoprotein receptor-binding domain (RBD) and to the peptidase (PD)–ACE2 domain, which are necessary for the entry of the virus and for the latter to produce a viral infection [46].

Lastly, it has been suggested, in some investigations on curcumin, that, in addition to its being effective against COVID-19, it is also effective against diabetes in terms of controlling the blood glucose, improving the function of the cells, preventing the necrosis or apoptosis of the islets of Langerhans, diminishing insulin resistance, and helping in the prevention of the damage wrought by diabetes-associated complications such as diabetic nephropathy and cardiopathy, due to curcumin's anti-inflammatory and antioxidant properties [47,48]. Other studies have found that the effect of curcumin on diabetes is related to the increase of the peroxisime proliferator-activated receptor (PPAR) and its effect on the increase of the gene expression of GLUT4, GLUT2, and GLUT3, and due to the expression of the adiponectin genes [49]. Moreover, it has been observed that curcumin, in combination with metformin, diminishes glyco-oxidative stress in diabetic rats [50]. Chuengsamarn et al. [51] reported that the intervention of curcumin in the functional improvement of β-cells was associated with the increase in HOMA-b and the reduction of the C-peptide.

The information collected on the potential effects of curcumin justifies it being catalogued as a promising coadjuvant for its utilization.

### 4.2. Silymarin

Silymarin, an extract from the seeds of the milk thistle (*Silybum marianum*), of Mediterranean origin, contains a gamma of flavolignans such as silybin, isosilybin, silychristin, and silydianin [52]. Silymarin has been credited with having effects on liver disorders, in addition to anticancer, antihepatoprotector, antihyperlipidemic, antidiabetic, antiamnesic, anti-inflammatory, and antisclerotic activities, and it modulates T cells [53–55]. It has been considered effective for the symptoms of respiratory diseases, fever, and influenza [56]. Zhu, Z and Sun, G. [57] referred to silymarin as effective in alleviating lung injury, due to the fact that it inhibits inflammation in the tissue, even blocking ARDS development.

Accordingly, the characteristics of silymarin have had beneficial effects on several pathologies. However, some studies have suggested that its potential is limited due to poor intestinal absorption and low bioavailability, with 23–47% in systemic circulation followed by oral administration [58–61]. For this reason, there has been a suggestion for formulations research to increase the bioavailability, such as: soft capsules, nanoparticle, liposomes [61–63].

### Silymarin and Its Effect on COVID-19 and Diabetes

Gorla et al. [64] have considered silymarin as a potent ACE-2-peptidase domain (PD)-ACE-2) inhibitor, due to the fact that silymarin maintains coupling with the active site. Saraswat et al. [65] found that silymarin maintained binding with the principal protease of SARS-CoV-2; thus, it could be utilized as an antiviral agent. Another study demonstrated the binding of diverse phytochemicals, among these the silymarin of the viral protein, nonstructured protein 15 (Nsp15), which inhibited the replication of SARS-CoV-2 [66].

The silybin component of silymarin has a possible direct effect on the STAT3 regulator of the inflammatory cytokine pathway and on the immune response. It could be phenotypically related to the mechanisms of action of the monoclonal antibodies of IL-6 and pan-JAK1/2 inhibitors, with the purpose of inhibiting the production of cytokines and lymphopenia in T cells, both of which are recurrent in severe cases of COVID-19. In addition, it could be the most propitious inhibitor against SARS-CoV-2, due to its coupling with the target genes of the virus, inhibiting RNA-dependent RNA polymerase-mediated RNA (RdRp), which is indispensable in the replication/transcription of SARS-CoV-2 and above the peak of the ACE2 glycoprotein [67,68] (Figure 3).

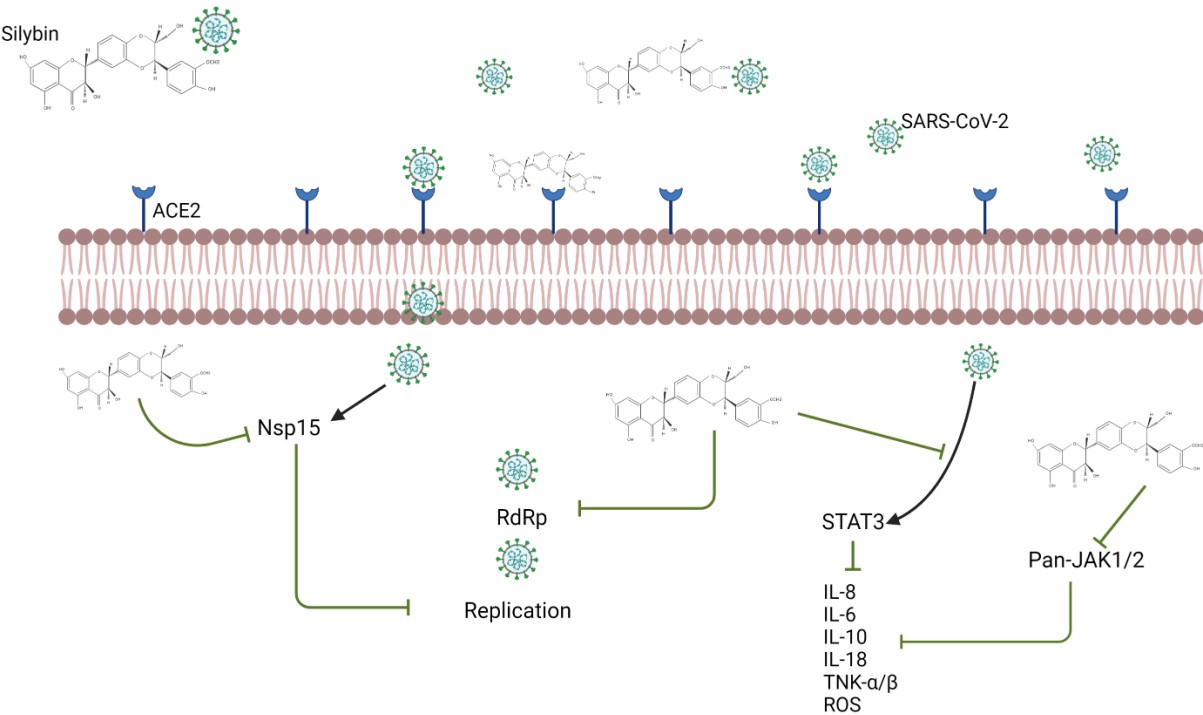

**Figure 3.** Mechanism of action of silymarin (silybin). The possible mechanism by which silymarin (silybin) interferes in the development of COVID-19 is due to its coupling with the peptidase–ACE2 domain of SARS-CoV-2 or Mpro, rendering binding in ACE impossible. Additionally, it could bind Nsp15, blocking the replication of SARS-CoV-2. On the other hand, silybin, the principal component of silymarin, could block the proinflammatory cytokine storm (specifically at IL-6) and T-cell lymphopenia. Created with BioRender.com (accessed on 1 September 2021).

In addition to exerting a possible effect against SARS-CoV-2, silymarin has been reported to exert an effect on diabetes, increasing serum insulin and the neogenesis of cells, normalizing serum glucose, and avoiding the development of diabetic complications. The effect is attributable to the antioxidant action that silymarin maintains, such as the following: inhibiting the formation of free radicals, reducing the inflammatory responses as a consequence of the inhibition of NF-kB-dependent pathways, and maintaining optical redox equilibrium due to the activation of antioxidant enzymes and nonenzymatic antioxidants by means of the activation of Nrf2 [69]. Mononmani et al. [70] showed that silymarin maintains effective glucose control in diabetic patients with hepatic lesions, suggesting that the effect is associated with the antioxidant activity that exerts an impact, diminishing insulin resistance. Stolf et al. [69] noted that silymarin and/or its components, in confronting diabetes, have the capacity to attenuate the usual diabetic complications in diverse organs. Glucose levels diminished in rats after the administration of silymarin, improving insulin levels, due to a restoration of the pancreatic β-cells [71]. It has been observed that metformin, silymarin, and angiotensin–renin or angiotensin receptor blockers, in addition to maintaining the glucose (metformin), can also present a preventive or delaying effect on the progression of diabetic nephropathy, connected to the antioxidant

and anti-inflammatory silymarin effects [72]. Additionally, it has been reported that, in patients with type 2 diabetes plus nephropathy, there is a marked reduction of the urinary excretion of albumin, TNF-α, and malondialdehyde when silymarin is administered in combination with renin–angiotensin inhibitors [73].

### 4.3. Sulforaphane

Sulforaphane is an isothiocyanate deriving from the hydrolysis of glucosinolate glucoraphanin (GRA) in cruciferous vegetables such as Brussels sprouts, cabbages, and, mainly, in broccoli [74,75]. Studies have shown that sulforaphane exerts effects on diseases including neurological diseases and chronic pulmonary diseases, and antidiabetic, anticancer, and anti-inflammatory effects [74–80].

These benefits are due to sulforaphane having faster absorption and high bioavailability [81,82], with appropriate plasma and urine (low excretion) concentrations present [83].

Sulforaphane and Its Effect on COVID-19 and Diabetes

It has been observed that, in patients with infection by COVID-19, their condition often becomes critical; therefore, the proinflammatory system (C-reactive protein, the cytokines IL-6 and IL-8, and TNF-α, etc.) is activated. There is an increase of ROS that gives rise to cellular and tissue damage. The acute respiratory distress syndrome (ARDS) develops and there is an alteration in vascular permeability (endothelial dysfunction and thrombosis). It has been suggested that the activation of Nfr2 could improve the state caused by SARS-CoV-2; to date it has suppressed the expression of cytokines and activation of the inflammasome; in addition, it protects the respiratory epithelium and the vascular endothelium and enhances the expression of antioxidant enzymes [84]. It has been reported that sulforaphane tends to be an activator of Nrf2. On the other hand, sulforaphane has been proposed as a strategic treatment against COVID-19, because this compound possesses the function of activating Nrf2, which provides cytoprotection due to the homeostasis of proteins and redox, in addition to it inhibiting inflammation [85]. According to Benarba et al. [86], natural products such as sulforaphane maintain a beneficial effect on confronting SARS-CoV-2 in terms of blocking the ACE2 receptor or the serine protease TMPRRS2, which are indispensable for human cellular infection (Figure 4).

According to studies carried out by Axelsson et al. [87], the authors reported that diabetes treated with sulforaphane inhibits the production of glucose by hepatic cells through the nuclear translocation factor 2, relating it to erythroid nuclear factor 2 (Nrf2), and that, in addition, it produces a diminution in the key enzymatic expression of glyconeogenesis and attenuates glucose intolerance. It was also observed that sulforaphane reduces fasting blood glucose and glycosylated hemoglobin (HbA1c) in obese populations with uncontrolled type 2 diabetes. Patel et al. [88] suggested sulforaphane as a potent activator of Nrf2, whose effect is to revert the diabetic, cardiometabolic, and cytoprotective complications related to proinflammatory factors such as Nf-kB PPAR. The findings reported by Lv et al. [89] suggest that sulforaphane could delay diabetic retinopathy as a result of the degeneration of the photoreceptors in diabetes and sulforaphane may inhibit stress in the endoplasmatic reticule and the inflammation and expression of Txnip by activation of the AMPK pathway.

In COVID-19, diabetes predisposes to the incidence of COVID-19, increases disease severity, and even causes COVID-related death [90]. This is because SARS-CoV-2 could be involved in chronic inflammation, in the increase of the activity of coagulation, and in the destruction of the immune response, as well as in possible pancreatic injury, thus associating itself with diabetes and COVID-19 [91]. Therefore, it is important to find therapeutic alternatives destined for the control of diabetes and COVID-19, as well as the use of sulforaphane, which has been described as maintaining an effective control over diabetes, but that additionally has been considered to manifest an antiviral effect.

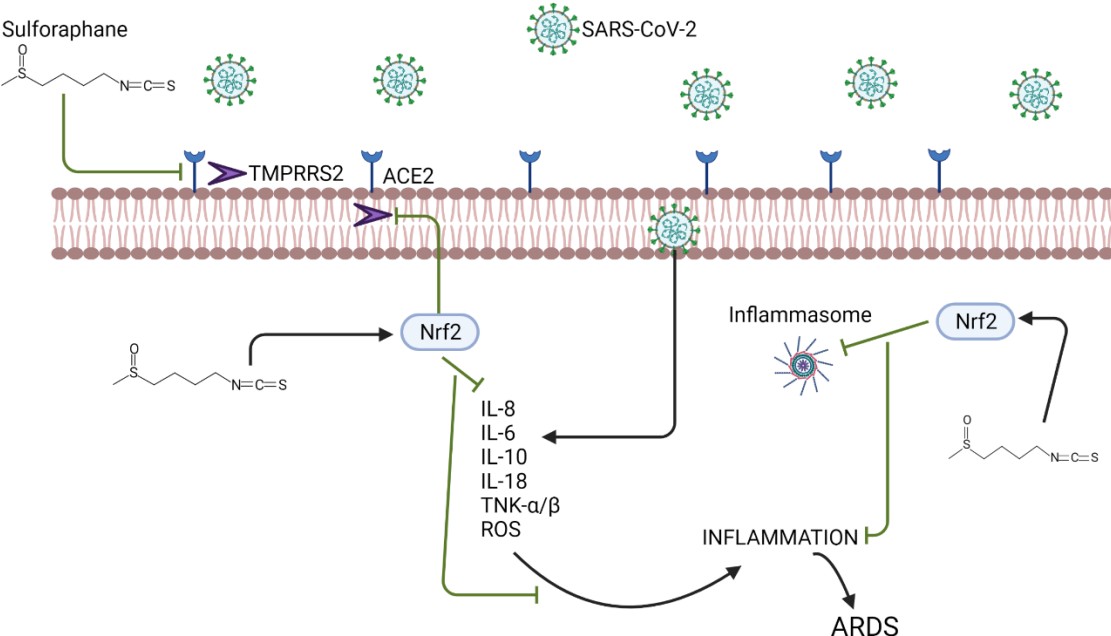

**Figure 4.** Possible mechanism of action of sulforaphane on COVID-19. Sulforaphane is related to the ACE2 protein and inhibits its binding with SARS-CoV-2. Additionally, sulforaphane binds with the TMPRRS2 serine, impeding SARS-CoV-2 infection. Sulforaphane activates Nrf2 and blocks the expression of proinflammatory molecules. Nrf2 inhibits the expression inflammasome. Created with BioRender.com (accessed on 1 September 2021).

## 5. Conclusions

Uncontrolled or poorly treated diabetes renders the person with the disease at greater risk for contracting chronic damage to the organs, as well as for infections such as the viral ones. Hundreds of infections and deaths are caused by SARS-CoV-2 (COVID-19), and this is increasing even more so due to the underlying risk factors possessed by some populations, such as those with diabetes. Therefore, a variety of strategies have been implemented for the benefit of this population. Among these strategies, we reviewed the use of phytochemicals such as curcumin, silymarin, and sulforaphane, which have shown to be effective against COVID-19, but that, in addition, could maintain a benefit against other diseases. However, it is important to continue to conduct in vitro, in vivo, and clinical investigations with the purpose of clarifying the mechanisms of action of each of these reported phytochemical compounds.

**Author Contributions:** Conceptualization and original figures preparation: E.M.S.-G., writing—original draft preparation: D.E.-L., J.J.C.-C., R.C.J.-S., O.R.F.-C., R.M.B.-T.; writing—review and editing: E.R.-M., N.V.-M., J.A.-R., J.A.M.-G. All authors have read and agreed to the published version of the manuscript.

**Funding:** This research received no external funding.

**Institutional Review Board Statement:** Not applicable.

**Informed Consent Statement:** Not applicable.

**Data Availability Statement:** Not applicable.

**Acknowledgments:** The authors acknowledge the support of the concession of a doctoral fellowship Consejo Nacional de Ciencia y Tecnología: CONACyT 732974 for Eli Mireya Sandoval Gallegos. The authors thank Marcelo Angeles Valencia for the realization of the figures.

**Conflicts of Interest:** The authors declare no conflict of interest.

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
