# Peer review of "Phytochemicals and Their Possible Mechanisms in Managing COVID-19 and Diabetes"

_applsci, doi:10.3390/app11178163_

Round 1

Reviewer 1 Report

The review covers important information concerning possible use of phytochemicals for managing COVID-19 and diabetes. I have following recommendations: data about how the presented literature search was performed (sources, covered period of time, key word for searches etc) should be given, scientific names of plant species (lines 159 and 224) should be written in italisc.

Author Response

Reviewer 1:

The review covers important information concerning possible use of phytochemicals for managing COVID-19 and diabetes.

A) I have following recommendations: data about how the presented literature search was performed (sources, covered period of time, key word for searches etc.)

Answer: Thank you for the suggestion.  We add information requested in the summary (line 21-23), as well as introduction (line 57-61).

B) should be given, scientific names of plant species (lines 159 and 224) should be written in italisc.

Answer: Thank you for the suggestion. We have modified the words now are visible on lines 175-176and 250

Reviewer 2 Report

The article is extremely interesting, having in mind the importance of Covid-19 managment for all population, but especially for those with chronic diseases such as diabetes.

The review is well written and understandable, still I have some recomendations for the authors:

  1. why did You choose these 3 bioactive compounds (sulphoraphane, silymarin and curcumin), since there are also other phytochemicals with hypoglycemic and antiviral activity (such as thymoquinone, epigallocatechin gallate, isoflavones etc). The authors should clearly state an explanation in the article
  2. line 159 - curcumin is the phenolic compound, turmeric is the rhizome of Curcuma longa that contains this polyphenol.
  3. line 230 - I think it is silymarin instead of curcumin
  4. I suggest the authors to include a brief paragraph regarding the bioavailability of the mentioned compounds. Although the results are promising there is a great difference between in vitro and in vivo activity (for example curcumin although it shows a wide range of therapeutic effects it has a very low bioavailability)

Author Response

Reviewer 2:

The article is extremely interesting, having in mind the importance of Covid-19 management for all population, but especially for those with chronic diseases such as diabetes. The review is well written and understandable, still I have some recommendations for the authors:

1. why did You choose these 3 bioactive compounds (sulphoraphane, silymarin and curcumin), since there are also other phytochemicals with hypoglycemic and antiviral activity (such as thymoquinone, epigallocatechin gallate, isoflavones etc). The authors should clearly state an explanation in the article

Answer: Thanks for your question. We add explanation among the line 167 and 172.  There are several compounds that have been identified with hypoglycemic and antiviral activity, However, the phytochemicals curcumin, silymarin and sulforaphane have been widely used in worldpopulation eighter as elements of the diet or in traditional medicine to treat medical conditions. Recently, these compounds had taken scientific attention because of the different properties described such anti-inflammatory, antioxidant, antimicrobial, hepatoprotective, among others.

2- line 159 - curcumin is the phenolic compound, turmeric is the rhizome of Curcuma longa that contains this polyphenol.

Answer: Thank for your observation. Exactly the curcumin is obtained of the rhizome of the curcuma plant. Is visible on line 175, this was modified.

3. line 230 - I think it is silymarin instead of curcumin.

Answer: Thank for your observation. The description was clarified in both compounds (lines184-186 curcumin and lines 256-257 silymarin)

4. I suggest the authors to include a brief paragraph regarding the bioavailability of the mentioned compounds. Although the results are promising there is a great difference between in vitro and in vivo activity (for example curcumin although it shows a wide range of therapeutic effects it has a very low bioavailability)

Answer: Thank for your suggestion. We add bioavailability information in each phytochemical compound (curcumin lines 187-193; silymarin lines259-263 and sulforaphane line 309-310).